# Dietary-Induced Ketogenesis: Adults Are Not Children

**DOI:** 10.3390/nu13093093

**Published:** 2021-09-02

**Authors:** Keren Porper, Leor Zach, Yael Shpatz, Bruria Ben-Zeev, Michal Tzadok, Elisheva Jan, Alisa Talianski, Colin E. Champ, Zvi Symon, Yair Anikster, Yaacov R. Lawrence

**Affiliations:** 1Pediatric Neurology Institute, Edmond and Lily Safra Children’s Hospital, Sheba Medical Center, Tel HaShomer, Ramat Gan 52621, Israel; kerenporper83@gmail.com (K.P.); Bruria.BenZeev@sheba.health.gov.il (B.B.-Z.); Michal.Tzadok@sheba.health.gov.il (M.T.); 2Institute of Oncology, Sheba Medical Center, Tel HaShomer, Ramat Gan 52621, Israel; Alisa.Talianski@sheba.health.gov.il; 3Sackler School of Medicine, Tel Aviv University, Tel-Aviv 6997801, Israel; leor.zach@sheba.health.gov.il (L.Z.); Zvi.Symon@sheba.health.gov.il (Z.S.); Yair.Anikster@sheba.health.gov.il (Y.A.); 4Department of Radiation Oncology, Sheba Medical Center, Tel HaShomer, Ramat Gan 52621, Israel; Yael.Shpatz@sheba.health.gov.il (Y.S.); Elisheva.Jan@sheba.health.gov.il (E.J.); 5Department of Radiation Oncology, Duke University, Durham, NC 27710, USA; colinchamp@tutanota.com; 6Metabolic Diseases Unit, Edmond and Lily Safra Children’s Hospital, Sheba Medical Center, Tel HaShomer, Ramat Gan 52621, Israel; 7Department of Radiation Oncology, Sidney Kimmel Medical College, Thomas Jefferson University, Philadelphia, PA 19107, USA

**Keywords:** ketogenic diet, ketogenesis, glioblastoma, epilepsy, age-related differences

## Abstract

There is increasing interest in the use of a ketogenic diet for various adult disorders; however, the ability of adults to generate ketones is unknown. Our goal was to challenge the hypothesis that there would be no difference between adults and children regarding their ability to enter ketosis. Methods: Two populations were studied, both treated with identical very low-carbohydrate high-fat diets: a retrospective series of children with epilepsy or/and metabolic disorders (2009–2016) and a prospective clinical trial of adults with glioblastoma. Dietary intake was assessed based upon written food diaries and 24-h dietary recall. Ketogenic ratio was calculated according to [grams of fat consumed]/[grams of carbohydrate and protein consumed]. Ketone levels (β-hydroxybutyrate) were measured in blood and/or urine. Results: A total of 168 encounters amongst 28 individuals were analyzed. Amongst both children and adults, ketone levels correlated with nutritional ketogenic ratio; however, the absolute ketone levels in adults were approximately one quarter of those seen in children. This difference was highly significant in a multivariate linear regression model, *p* < 0.0001. Conclusions: For diets with comparable ketogenic ratios, adults have lower blood ketone levels than children; consequently, high levels of nutritional ketosis are unobtainable in adults.

## 1. Introduction

The ketogenic diet (KD), consisting of high fat, moderate to low protein, and very low carbohydrate intake, has proven efficacy in children with refractory epilepsy [1,2], and is standard of care treatment for certain genetic metabolic disorders [3,4]. In adults, there is increasing use of KDs in drug-resistant epilepsy, even though their therapeutic impact is uncertain [5]. Furthermore, there is recent interest in the use of a KD in the treatment of various adult disorders including cancer [6], diabetes mellitus, and obesity [7]; ongoing clinical trials are summarized in Appendix A. Whereas in children the ability of a low-carbohydrate high-fat diet to induce systemic ketosis has been demonstrated in multiple prospective studies [8,9], the adult data are far more limited; it is unknown whether adults and children stimulate diet-induced systemic ketosis to the same extent, or what factors promote ketosis in the adult population. This is especially important since, at least in children, blood ketone levels correlate with efficacy—seizure control—in the epileptic population [8,9].

We recently performed a prospective clinical trial in which a very low carbohydrate high-fat diet was administered to adults with brain tumors [10]. Despite the input of an experienced ketogenic-nutrition team, maintenance of nutritional ketosis proved surprisingly difficult in this population. Our null hypothesis was that there is no difference in dietary-induced systemic ketosis. Here we perform a detailed dietary analysis of two populations: adults with recurrent glioma who participated in the clinical trial and children with refractory epilepsy or metabolic disorders, seeking to correlate nutritional intake with ketone body levels.

## 2. Materials and Methods

The investigation comprised two different study patient populations both within the Sheba Medical Center:
(1)A retrospective sample of children treated with a long-term KD in the pediatric neurological department during the years 2009–2016 for either epilepsy or metabolic disorders: GLUT1 deficiency syndrome (GLUT1DS) and pyruvate dehydrogenase deficiency (PDHD). Inclusion criteria were children who received the KD between the ages of six months to 18 years diagnosed with epilepsy who did not respond to at least two types of medication, with a frequency of seizures at least once a day or at least seven seizures per week; treatment with the KD was for over three months. Exclusion criteria included children with refractory epilepsy who had received treatment with KD for less than three months; patients for whom there was a lack of relevant information from the medical records; and children with known metabolic disease associated with lipid metabolism.(2)A prospective single-institution clinical trial (NCT02149459) [10] amongst adults with newly diagnosed or recurrent high-grade glioma, in which patients received the KD (for a total of eight weeks) prior to, during, and following two or six weeks of radiation therapy. Patients in cohort one received dietary treatment alone, whereas those in cohorts two and three received metformin in addition. Inclusion criteria were patients above the age of 18 years suffering from high-grade glioma (brain tumor). Exclusion criteria included patients with known metabolic disease associated with lipid metabolism, diabetes mellitus, and known severe dyslipidemia.

### 2.1. Dietary Intervention

Subjects were instructed to conform to either the classical KD (ketogenic ratio between 4:1 and 3:1) [11,12] or modified Atkins diet (MAD) (estimated ketogenic ratio between 1:1 and 3:1) [13]). (Dietary details are presented in Appendix A). Medium chain triglyceride (MCT) coconut-extracted oil thought to stimulate hepatic production of β-hydroxybutyrate (β-OHB) ketone bodies [14,15] was added at the dietician’s discretion.

Subjects (and parents for pediatric subjects) met with a dietitian on a regular basis prior to and during the KD in accordance international guidelines [1]. Dietary intake was evaluated by means of detailed daily food diaries or 24-h dietary recall. Daily nutritional analysis was calculated using Tzameret software (Maymone Software, Jerusalem, Israel).

### 2.2. Ketogenic Ratio

Ketogenic ratio was calculated according to the formula: total fat intake (gram) divided by the sum of total protein intake (gram) and total carbohydrate intake (gram) [11,12].

### 2.3. Ketone Bodies

β-OHB ketone bodies were measured either in venous blood (hospital laboratory or bedside testing using the Precision Xtra Ketone Monitoring System (Abbott Laboratories, Lake Bluff, IL, USA)) and/or in urine using a dipstick. For those pediatric visits in which only urinary ketones levels were available, venous ketone levels were estimated using the formula and analysis presented in Appendix A. Ketones levels in adults prior to commencing the diet were considered to be zero.

### 2.4. Statistical Methods

The dietary components were compared prior and during the diet intervention using Student’s *t* test. The relationship between blood ketone levels and dietary intake, anthropometric measurements, and population type (children vs. adults) was examined using a univariate and multivariate liner mixed model. Correlations between ketogenic ratio and blood ketones level were graphically demonstrated using a two-way quadratic prediction plot. The differences between children (aged less than 18 years) and adults (aged 18 years or more) were tested using a multivariate linear regression model, treating adult status as a binary variable. Statistical analysis was performed using the Stata statistical package, version IC 11.1 (Stata, College Station, TX, USA).

### 2.5. Ethical Consideration

Both the prospective clinical trial in adults and the retrospective study in children were approved by the Sheba institutional review board (IRB). The prospective clinical trial is listed on Clinicaltrials.gov, NCT02149459.

## 3. Results

### 3.1. Subject’s Characteristics

The study included 15 children and 13 adult patients. A summary of baseline demographic characteristics is presented in Table 1. (Additional demographic details are presented in Appendix A). A total of 168 observations were made amongst children and adults.

### 3.2. Dietary Intake Prior and during Diet Intervention

Both populations were compliant with the diet, as assessed by diet recall and detailed diet diaries, and confirmed by a significant decrease in the carbohydrate contribution to daily caloric intake in both children and adult populations three weeks after diet initiation (from 51.7 ± 8.2% to 4.1 ± 2.0%, *p* < 0.001, from 30.5 ± 3.7% to 5.7 ± 0.5%, *p* = 0.0001, respectively). Conversely, the fat contribution to daily caloric intake increased significantly (from 35.8 ± 6.5% to 88.7 ± 2.3%, *p* < 0.001, from 49.2 ± 15.5% to 73.8 ± 9.4%, *p* = 0.0025, respectively). (Dietary details are presented in Appendix A).

### 3.3. Anthropometric and Metabolic Parameters

During the dietary intervention, there was a significant increase in systemic (blood/serum) ketone levels (β-OHB) in both children and adult populations, whereas there was no significant change in anthropometric and other metabolic parameters (Appendix A). During the dietary intervention, systemic (blood/serum) ketone levels (β-OHB) were significantly higher in children as compared with adults (3.4 ± 1.5 mmol/L vs. 0.6 ± 0.67 mmol/L, respectively, *p* < 0. 001).

### 3.4. Associations with Blood Ketone Levels

On univariate analysis including both children and adults, dietary intake including total calories per kg of body weight, fat intake (as a percentage contribution to total caloric intake), lower protein intake (as a percentage contribution to total caloric intake), lower carbohydrate intake (as a percentage contribution to total caloric intake), MCT intake and the ketogenic ratio, population type (children vs. adults), and anthropometric measurements, including lower weight and BMI, were significantly correlated with higher blood ketone levels (Table 2).

### 3.5. Children versus Adults

The association between ketogenic ratio and blood ketone levels was noticeably different between children and adults, with the latter having far lower ketone levels (Figure 1).

### 3.6. Multivariant Mixed-Model Analysis

The multivariant mixed-model analysis included the following covariates: total calories per kg of body weight, fat intake (as a percentage contribution to total caloric intake), carbohydrate intake (as a percentage contribution to total caloric intake), MCT per kg of body weight and population type (adults = 0 vs. children = 1). The only covariates significantly associated with ketone levels were fat-intake and population type (Table 3).

## 4. Discussion

We demonstrate that the induction of systemic ketosis is highly dependent on dietary intake. Nutritional ketosis in both children and adults only occurred when the diet contained high levels of fat with very little carbohydrate. Furthermore, we demonstrate that for the same ketogenic ratio, circulating ketone-body levels in adults are far lower than in children.

An important novel finding in our study is that for the equivalent ketogenic ratio, adults had much lower blood ketone levels than children. An association between age and diet-induced ketogenesis in humans has not previously been reported; however, when reviewing the overall literature, it appears that while ketogenesis (2–4 mmol/L) is achievable in children [8,9,16], researchers appear to have great difficulty achieving similar levels in adults. We identified ten prospective published studies that have investigated the use of KD in adults (Appendix A). Several of the studies did not report ketones levels in detail; those that did reported only low to moderate levels of ketosis (0.5–1.5 mmol/L). Likewise, Hennebelle et al. reported a study of the influence of linoleic fat supplementation on fasting-induced ketosis. They demonstrated that younger adults (mean age 25 years) were more prone to ketosis than older adults (mean 73 years) [17].

Our findings are supported by studies in animals (Appendix A); young, but not older rats, achieved a therapeutic level of ketones (2–4 mmol/L) when fed a KD (ketogenic ratio 4:1) [18]. Likewise, amongst rats fed a calorie-restricted KD, the younger animals generated significantly higher β-OHB ketone levels than the older animals [19]. Another study of caloric fasting (36 h) in rats demonstrated an age-dependent difference in acetoacetate (AcAc) ketone levels [20]. Of note, in our study, higher total calories per kg of body weight were associated with higher ketone levels on the univariate analysis. This could be related to the fact that children weigh less and need more calories per kg for growth compared to adults, rather than the effect of restricted total calories compared to unlimited total caloric input on ketone levels.

The biochemical mechanism underlying age-dependent differences in dietary-induced ketosis is unknown. Possible reasons include higher postprandial insulin levels frequently seen in older adults, lower plasma carnitine levels, and lower plasma lipoprotein lipase activity in older adults [20,21,22,23]. The rate of hepatic ketogenesis is known to be regulated by the supply of free fatty acid (FFA) [24] and by the hormones insulin and glucagon [25]. Insulin inhibits FFA release from adipose tissue and, hence, ketogenesis [26]; likewise, in our adult cohort, we noted a negative correlation between insulin levels and ketone levels (*p* = 0.02) [10]. The higher postprandial insulin response seen in older people [17] would hence help explain their difficulty in achieving systemic ketosis. Conversely, glucagon stimulates hepatic ketone body production in liver; rat studies have suggested that adult livers are less sensitive to glucagon than young animals’ livers [20].

Additionally, changes in availability of FFA (essential for ketone body production), may explain age-related difference in ketosis [27]. Lipoprotein lipase is responsible for FFA release from triglyceride; in human plasma lipoprotein lipase activity decreases with age [23]. Carnitine is essential for the entry of FFA into the mitochondria where ketogenesis occurs, with age plasma carnitine levels decrease [20]; hence age-related changes in lipoprotein lipase activity and carnitine levels may also explain the relatively impaired ketone production in the adults compared with children. An extreme demonstration of age-dependent changes in dietary ketosis is the frequent ketotic episodes experienced by breastfeeding infants [28,29,30,31].

Strengths of our study include the assessment of two different populations within the same medical center, treated by the same dietician using similar guidelines and interventions. Dietary intake was closely monitored in all populations, including regular telephone contact. This is the first study, to our knowledge, directly comparing two groups of adults and children, and their ability to engage in nutritional ketosis. As more than 100 clinical studies utilizing a KD are currently underway (clinicaltrials.gov), this information is important and will help in future clinical trial design.

Our study has a number of limitations: the small sample size, this was partially overcome through the use of repeated observations of the same subjects, and a mixed models approach to data analysis. The two populations differed not only in age, but also regarding their comorbidities: whereas the children had either refractory epilepsy or metabolic disorders, the adult participants uniformly suffered from brain tumors. It is possible these medical conditions and their respective medications may have influenced dietary ketogenesis. Another limitation in our study is the different types of KD that were used in the two populations. Adult patients were all treated with MAD, which contains considerable quantities of protein, while the children were mostly treated with a protein-depleted classic KD. We calculated the ketogenic ratio based upon each individual’s intake in order to overcome dietary differences; however, this may be an over simplification since it gives equal weight to both carbohydrates and proteins, whereas in reality some amino acids stimulate gluconeogenesis—and are therefore detrimental to ketogenesis. The retrospective nature of the pediatric study was associated with some missing data, e.g., dietary intake prior to intervention, ketone levels measured in urine rather than in peripheral blood, and blood insulin levels. Furthermore, ketone levels were measured at different times of day but generally in the morning; there is some evidence that ketone levels rise and fall over the course of the day [32]. Despite these reservations, it is unlikely that these potential biases would fully explain the profound differences in ketosis noted in Figure 1.

## 5. Conclusions

We demonstrate that achieving high levels of nutritional ketosis is achievable in children but not in adults. This has not been previously appreciated, and the mechanisms underlying this difference deserve investigation. Future studies should seek to validate our findings, while examining additional factors that may influence dietary ketosis, including anthropometric measurements, physical activity, and the microbiome [33,34,35]. Ideally, these future studies should compare adult and pediatric populations with similar morbidities and dietary interventions, with age being the only distinguishing factor.

A more complete understanding of why adults are less prone to dietary-induced ketosis compared to children may facilitate a more efficient means of inducing therapeutic ketosis in all populations.

## Figures and Tables

**Figure 1 nutrients-13-03093-f001:**
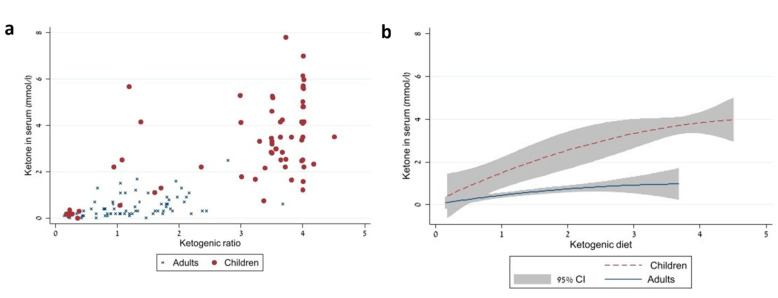
Correlation between dietary ketogenic ratio and blood ketone levels in the two populations: children vs. adults (*p* < 0.0001). The raw data are presented in panel (**a**), quadratic modeling is presented in panel (**b**).

**Table 1 nutrients-13-03093-t001:** Baseline demographics of participants.

Parameter	Children (*n* = 15)	Adults (*n* = 13)
Sex:		
Male	7 (47%)	8 (61.5%)
Female	8 (53%)	5 (38.5%)
Mean age (years), SD	3.0 ± 3.3	61.4 ± 5.3
Diagnosis:		
Epilepsy	13 (87%)	-
Metabolic disorder	2 (13%)	-
Brain tumor	-	13 (100%)
Diet Type:		
Classical ketogenic diet	11 (73%)	
Modified Atkins diet	1 (7%)	13 (100%)
Combined	3 (20%)	

**Table 2 nutrients-13-03093-t002:** Covariate associations with blood ketone levels, mixed-model univariate analysis.

Variable	Coefficient	*p*-Value	CI 95%
Fat intake (%)	0.05	<0.0001	0.04, 0.06
Carbohydrate intake (%)	−0.05	<0.0001	−0.067, −0.04
Protein intake (%)	−0.09	<0.0001	−0.12, 0.06
Calories (kcal) per kg	0.30	<0.0001	0.02, 0.04
Ketogenic ratio	0.93	<0.0001	0.78, 1.08
MCT (ml) per kg	0.86	0.016	0.15, 1.5
Weight (kg)	−0.03	<0.0001	−0.04, −0.02
BMI	−0.16	<0.0001	−0.21, −0.11
Population type, (Adults = 0, Children = 1)	2.62	<0.0001	2.09, 3.15

**Table 3 nutrients-13-03093-t003:** Mixed-model multivariate analysis, covariates associated with blood ketone levels.

Variable	Coefficient	*p*-Value	CI 95%
Fat intake (%)	0.05	0.006	0.014, 0.086
Carbohydrate intake (%)	0.04	0.20	−0.02, 0.11
Total calories (kcal per kg)	0.004	0.52	−0.009, 0.018
MCT (mL per kg)	−0.42	0.18	−1.03, 0.20
Population type (Adults = 0, Children = 1)	1.7	0.001	0.7, 2.68

## Data Availability

The data that support the findings of this study are available on request from the corresponding author (Y.R.L.) in accordance with IRB permissions.

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
