# Peer review of "Dietary-Induced Ketogenesis: Adults Are Not Children"

_nutrients, 2021, doi:10.3390/nu13093093_

Round 1

Reviewer 1 Report

The current manuscript investigated whether the extents of dietary-induced ketogenesis are different between adults and children. The research topic is interesting and meaningful for related research. The study was well designed, data and results were clearly presented, and discussion and information in supplementary materials were useful.

Introduction

Introduce and emphasize why it is important to investigate whether adults and children induce diet-induced systemic ketosis to a similar extent or not.

Materials and Methods

Lines 59, 60, 63, and 64: list types of metabolic disorders at appropriate locations.

Lines 94 and 107: check the journal guideline for supplier information.

Results

Table 1: check number for diet type in children.

Lines 128, 129, 161, and 166: check the units.

Include tables for demography, anthropometry, biochemical parameters in blood /urine, and dietary intakes.

Include serum concentrations of FFA and insulin if the authors have. Then, the authors can explain the possible mechanisms in lines 182-186 with the supporting data. 

Discission

Differences in medical conditions (type of diseases) may affect the extent of ketosis. Discuss this point.

Check/revise space, grammar, and typoes throughout the manuscript.

Reviewer 2 Report

In their study  Porper and coll. compared blood ketone bodies in 15 children with epilepsy or metabolic disorders and in 13 adults with glioblastoma, finding lower levels of ketonemia in adults. Despite the low sample size, the study was well conducted , with a thorough statistical analysis.  However, the study have some limitations beyond those described by the authors.

Children develop ketosis more easily than adults and this fact has long been known by pediatricians. Moreover, in this study there are two important differences between adults and children. First, adult patients were all treated with Atkins diet, richer in protein than the classic ketogenic diets, while the children were mostly treated with the classic 3:1 ketogenic diet. Dietary ketogenic ratio is a is a gross indicator of ketogenicity, as it adds lipids and proteins as if they had the same ketogenic potentiality. In reality, we know that proteins are much less ketogenic as many amino acids are gluconeogenic. To appropriately assess the differences, the two groups had to be treated with the same type of diet.

Second, metabolic alterations in the neoplastic patient (eg, stimulation of gluconeogenesis) could modify the response to the ketogenic diet. The fact that all adult patients were neoplastic, whereas none of the pediatric subjects were, might also have influenced the observed differences.

The authors should comment on these points in the discussion.

Round 2

Reviewer 2 Report

In my opinion the manuscript can be accepted for pubblication in present  form